# More than Just Host Plant Preferences for the Two Main Vectors of *Xylella fastidiosa* in Europe: Two Insect Species and Two Different Behaviors

**DOI:** 10.3390/insects16040416

**Published:** 2025-04-15

**Authors:** Saúl Bernat-Ponce, Rosalía García-García, Cristina M. Aure, Lorena Nieves, Juan Pedro Bouvet, Francisco J. Beitia, César Monzó

**Affiliations:** 1Centro de Protección Vegetal y Biotecnología, Instituto Valenciano de Investigaciones Agrarias, CV-315, Km 10.7, 46113 Valencia, Spain; berpon@alumni.uv.es (S.B.-P.); rosalia.garcia@uv.es (R.G.-G.); martinez_criaur@gva.es (C.M.A.); niecalo@uv.es (L.N.); beitia_fra@gva.es (F.J.B.); 2Estación Experimental Agropecuaria (EEA), Instituto Nacional de Tecnología Agropecuaria (INTA), Ruta Nacional 14, Km 259, Concordia E3201, Entre Ríos, Argentina; bouvet.juan@inta.gob.ar

**Keywords:** *Philaeunus spumarius*, *Neophilaenus campestris*, host preference, insect behavior, Aphroporidae, oviposition, nymph development

## Abstract

*Xylella fastidiosa* is a plant disease-causing bacterium that has devastating impacts on global agriculture, affecting crops such as olives, grapes, and citrus. In Europe, two main insect species are responsible for transmitting the disease: the spittle bugs *Neophilaenus campestris* and *Philaenus spumarius*. Understanding how these insects interact with plants is essential to limiting the spread of the disease. In this work, we studied how plant species and habitat diversity influence the oviposition behavior of adult females and the development of nymphs in both species. *Neophilaenus campestris* showed a strong preference for grass plants for both egg-laying and nymph development, while *P. spumarius* primarily laid its eggs on dry soil substrates, regardless of the host plant species. Interestingly, *P. spumarius* females had increased oviposition rates when only a single preferential host was present. These findings underscore the importance of managing agricultural landscapes to disrupt the interactions between host plants and vectors, thereby reducing the spread of *X. fastidiosa*.

## 1. Introduction

*Xylella fastidiosa* (Wells et al., 1987) is a phytopathogenic bacterium that multiplies within the xylem of plants, forming biofilms that obstruct and impede sap flow [1]. With a broad host range exceeding 700 known plant species [2], *X. fastidiosa* is ranked among the 10 most important plant pathogens globally [3]. The bacterium is responsible for devastating plant diseases that impact economically important crops worldwide, including Pierce’s disease in grapevines, citrus variegated chlorosis in citrus, and olive quick decline syndrome in olive trees [4,5].

As a vector-borne pathogen, the transmission and spread of *X. fastidiosa* among plants are mediated by a diverse array of xylem sap-feeding insects characterized by high polyphagy [6]. The considerable diversity of potential insect vectors, coupled with the high genomic variability and plasticity of *X. fastidiosa* [5], results in highly complex pathosystems involving both cultivated and non-cultivated plant species. Among the key vector-related factors influencing disease dynamics are the abundance of insect vectors, their behavioral responses to environmental conditions and landscape features—such as the availability of preferential host plants [7]—their interactions with specific *X. fastidiosa* subspecies and clades [8], and their efficiency in transmitting the bacterium [9]. These vector-driven factors shape distinct pathosystems in affected regions, each requiring tailored management strategies [10,11,12,13].

All known vectors of *Xylella fastidiosa* belong to the suborder Auchenorrhyncha and infraorder Cicadomorpha and include members of the superfamilies Cercopoidea (families Aphrophoridae and Cercopidae) and Cicadoidea and the family Cicadellidae (subfamily Cicadellinae) [5,7,13]. In the Americas, the primary vectors belong to the subfamily Cicadellinae [13]. In contrast, in Europe, spittlebugs from the family Aphrophoridae are the only confirmed vectors of the bacterium [14]. Within this continent, transmission is mainly attributed to *Philaenus spumarius* (Linnaeus, 1758) and *Neophilaenus campestris* (Fallén, 1805) [15,16].

These two species are univoltine, completing a single generation per year [17,18,19]. Females lay eggs between September and December, depending on local environmental conditions, and the eggs overwinter until nymphs emerge from late February to late March. Upon hatching, nymphs settle on the nearest suitable green herbaceous host plant [15,20] and begin producing a protective foam—a defining characteristic of Aphrophoridae nymphs—that shields them from predators and environmental stressors. This foam formation is the reason they are commonly known as ‘spittlebugs’ [21,22,23]. Adults emerge in April and typically remain on the same vegetation.

In Mediterranean climates, the arrival of xeric summer conditions—characterized by reduced succulence of herbaceous plants, mowing, or drying—drives adults to migrate from the spring herbaceous hosts to the canopy of nearby crops, evergreen trees, or shrubs in search of shelter and protection [20,23,24,25,26]. This habitat shift is a key factor in the spread of *X. fastidiosa*, as adult insects may acquire the bacterium from infected plants and transmit it to healthy ones. At the end of summer, when xeric conditions subside, adults return to newly emerged green and succulent herbaceous plant habitats to complete their life cycle.

The host preferences of the two primary European vectors differ significantly. *Philaenus spumarius* is considered one of the most generalist herbivorous insects, being capable of feeding on over 1000 plant species, primarily from dicot families such as Asteraceae, Fabaceae, Apiaceae, Geraniaceae, and Asparagaceae [5,15,20,24,25]. In contrast, *Neophilaenus campestris* has a narrower host range, feeding predominantly on monocot plants, especially grasses and sedges in the family Poaceae [5,26].

The integration of non-crop plants into agroecosystems to enhance conservation biological control is increasingly adopted in Mediterranean agriculture [27,28]. However, in the context of *X. fastidiosa* pathosystems, this approach adds complexity to disease management. While increased plant diversity may heighten risks for crops, it also opens opportunities for managing vector populations through habitat manipulation.

Grass species such as *Festuca arundinacea* (Schreb) are frequently included in cover crops in Mediterranean orchards like citrus to support biological control of aphids or spider mites [29,30]. Legumes such as alfalfa (*Medicago sativa*) or sainfoin (*Onobrychis viciifolia*) are also integrated for their nitrogen-fixing abilities and their capacity to provide alternative food resources for natural enemies [31]. Additionally, the field marigold (*Calendula arvensis*) is a prominent component of winter agricultural and semi-natural Mediterranean landscapes [32].

Understanding the host selection strategies of *N. campestris* and *P. spumarius*—including their plant preferences, oviposition site selection, and responses to microhabitat features—could inform the development of sustainable vector management strategies through habitat manipulation, thereby limiting the bacterium’s spread.

This study aims to advance our knowledge of the field behavior of *P. spumarius* and *N. campestris* by investigating their interactions with plant species commonly found in Mediterranean agricultural habitats. We hypothesize that gravid females of both insect species can identify their preferential host plants irrespective of the habitat characteristics and will preferentially lay eggs near these plants. Additionally, we expect nymphs to settle directly on these hosts and develop successfully on them.

To test these hypotheses, we conducted three experiments: (1) analyzing the oviposition preferences of females on various plant species; (2) assessing nymphs’ ability to select suitable host plants for development; and (3) evaluating the nymphal development success on different host plants.

## 2. Materials and Methods

### 2.1. Insects and Plants

The nymphs and adults of *N. campestris* and *P. spumarius* used in all trials were obtained from continuous rearing maintained at the IVIA facilities. The rearing took place in a screenhouse divided into multiple separate cabins to facilitate different experiments. The environmental conditions in the screenhouse were semi-natural, as no artificial control of the temperature, relative humidity, or photoperiod was applied. Two plant species were selected for insect maintenance based on previous studies on nymphal presence, development, and adult feeding [13,14,15,16], as well as the author’s own experience. *Festuca arundinacea* L. (Poaceae), a grass species, was used to rear *N. campestris*, while *Medicago sativa* L. (Fabaceae), an herbaceous legume, was chosen for *P. spumarius*. Plants were grown from seeds in an insect-free greenhouse at the IVIA facilities. To maintain genetic diversity and population vigor, insect colonies were periodically supplemented with nymphs and adults collected from the field. Field collections were primarily conducted on herbaceous plants associated with olive groves and nearby seminatural habitats in the Valencia region of Spain. All experiments were conducted in the same screenhouse as the insect colonies but in separate experimental cabins.

### 2.2. Oviposition Patterns Across Vector Species

Experiments to investigate the oviposition patterns of each aphrophorid vector species were conducted during autumn and early winter, corresponding to the natural egg-laying periods of *N. campestris* and *P. spumarius* under field conditions. For each species, two temporal replicates were performed: December 2021 (mean temperature: 15.3 °C; mean RH: 58.6%) and November 2022 (mean temperature: 18.4 °C; mean RH: 75.4%) for *N. campestris* and October (mean temperature: 22.1 °C; mean RH: 67.0%) and November 2022 (mean temperature: 18.6 °C; mean RH: 73.2%) for *P. spumarius*. Three plant species commonly found in Mediterranean natural and agricultural habitats, which are frequently used as oviposition sites by aphrophorids, were selected for the experiments: field marigold (*C. arvensis* L.), alfalfa (*M. sativa*), and tall fescue (*F. arundinacea*). The temperature and relative humidity were continuously monitored throughout all of the experiments using dataloggers (Minidatalogger HD226-1, Delta Ohm SRL, Padova, Italy).

The experiments were conducted in entomological bugdorm tents (60 cm × 60 cm × 60 cm) located in one of the screenhouse cabins described earlier above. Each bugdorm served as a replicate. Two host diversity scenarios were tested: (1) a single-host scenario (no-choice oviposition), where only one host plant species was available, and (2) a multiple-host scenario (choice oviposition), where three different host plant species were present simultaneously. In each bugdorm, three potted plants were placed according to the assigned scenario. In the single-host scenario, all three potted plants belonged to the same species, while in the multiple-host scenario, one potted plant of each species (*C. arvensis*, *M. sativa*, and *F. arundinacea*) was included.

To simulate natural oviposition sites, five dry pine needles were placed on the soil surface of each pot and replaced daily. This set-up was based on previous studies and our own observations, which suggest that aphrophorid species, particularly *P. spumarius*, prefer to lay eggs on dry substrates such as pine needles [33]. Each bugdorm housed three females and five males collected from the IVIA colonies, and the females were allowed to oviposit for seven days. The pine needles were collected daily, transported to the laboratory, and examined under a binocular microscope to count the eggs. At the end of the experiment, the host plants were also analyzed under a microscope for the presence of eggs. For each potted plant, the total number of eggs deposited on the plant substrate and on the dry pine needles was recorded.

The number of replicates per temporal replicate and treatment was as follows. For *N. campestris* in December 2021, 4 and 8 replicates were used in the single-host and multiple host scenarios, and November 2022, there were 3 and 10 replicates, respectively. For *P. spumarius*, in October 2022, 4 and 8 replicates were used in the single-host and multiple-host scenarios, and in November 2022, there were 3 and 9 replicates, respectively.

### 2.3. Nymph Host Preferences

To assess whether newly emerged nymphs of the two aphrophorid species can detect and select an optimal host plant species for development, an experiment was conducted between March and April 2023 (mean temperature: 20.1 °C; mean RH: 53.3%) in the IVIA screenhouse. This period coincides with the natural emergence of *N. campestris* and *P. spumarius* nymphs in western Mediterranean habitats. The experimental set-up consisted of entomological bugdorm tents as described in previous sections. Within each bugdorm, a plant-growing plot 20 cm in diameter was placed, containing two *F. arundinacea* and two *M. sativa* plants. These plant species were selected based on their preferential host status: *F. arundinacea* for *N. campestris* and *M. sativa* for *P. spumarius*. The plants were grown in soil, with the two plants of the same species positioned together and spaced 16 cm apart from those of the other species.

For each replicate, five recently emerged first-instar nymphs of either *N. campestris* or *P. spumarius* were collected from the IVIA colonies and gently placed on the soil at the center of the plot equidistant (8 cm) from the nearest plant of each species. This set-up allowed the nymphs to move freely and select a host plant for settling and feeding. A total of 15 replicates (bugdorms) were conducted per insect species, testing 75 nymphs per species. The nymphs were monitored throughout their development, with minimal interference to avoid disrupting the protective foams they produced. The presence of nymphs on each plant species was recorded at specific time intervals: 15 min post-release (to assess initial host selection) and at 1, 2, 3, 8, 10, 17, 24, and 30 days after release (with the latter interval only measured for *P. spumarius*).

### 2.4. Nymph Survival and Development

The effects of host plant species on nymph development and survival were evaluated through separate experiments conducted in the screenhouse cabins. The experiment for *P. spumarius* was performed from March to May 2021 (mean temperature: 14.9 °C; mean RH: 68.3%), while the experiment for *N. campestris* took place from April to May 2022 (mean temperature: 19.5 °C; mean RH: 71.8%).

Four host plant species were tested in both experiments: *F. arundinacea*, a suitable host for *N. campestris*; *M. sativa* and *C. arvensis*, both known hosts for *P. spumarius*; and almond (*Prunus dulcis* (Mill.)), a woody crop that is unlikely to support aphrophorid nymph development but represents a key commercial crop in the *X. fastidiosa* pathosystem in Spain. For each experiment, two potted plants of the same species were placed inside each bugdorm, and five newly emerged first-instar nymphs from the IVIA colonies were released onto each plant (10 nymphs per replicate). Each plant species and insect species combination was tested in five replicates (bugdorms).

Nymph development was monitored twice weekly by recording the presence of protective foam on plants. To minimize disturbances, the foams were left intact. Upon reaching adulthood, the nymphs were removed, and the time elapsed from the start of the experiment to adult emergence was recorded for each individual. At the end of the experiment, the total number of individuals successfully developing into adults was counted for each bugdorm.

### 2.5. Statistical Analyses

Differences in oviposition rates between the vector species, host plant species, and host diversity scenarios were analyzed using a generalized mixed model with a negative binomial error distribution (Appendix A).

For each vector species, adult female oviposition preferences were assessed to determine how they varied with the host plant species and host diversity using generalized mixed models with a negative binomial error distribution (Appendix A).

To examine the host plant preferences of *P. spumarius* and *N. campestris* nymphs between monocotyledonous and dicotyledonous plants during their development, we used generalized mixed models with a Poisson error distribution (Appendix A).

For each observation time, the host plant preference differences for *P. spumarius* and *N. campestris* were analyzed using exact goodness-of-fit tests (Appendix A).

To determine whether the nymphal developmental time varied among the host plant species, we used a generalized mixed model with a Poisson error distribution (Appendix A).

Differences in *P. spumarius* nymphal survival rates across the host plant species (*C. arvensis*, *M. sativa*, and *P. dulcis*) were analyzed using a generalized mixed model with a binomial error distribution (Appendix A).

The Laplace approximation method was applied to estimate both random and fixed effects in all generalized mixed models. Post hoc Tukey’s tests were conducted to identify specific differences between treatment levels, with significance thresholds set at *p* < 0.05 and *p* < 0.10. All statistical analyses were performed using SAS^®^ OnDemand for Academics.

## 3. Results

### 3.1. Oviposition Patterns Across Vector Species

Both vector species displayed distinct oviposition strategies (Table 1). The mean number of eggs laid per plant during the seven days of the study was significantly higher for *N. campestris* (4.07 ± 0.55 eggs/plant) compared with *P. spumarius* (2.11 ± 0.45 eggs/plant) (vector: F = 6.20; df = 1, 419; *p* = 0.013).

The host plant species significantly influenced the oviposition rates (plant: F = 3.24; df = 2, 419; *p* = 0.040), with this effect varying between the two vector species (vector × plant: F = 7.4; df = 2, 419; *p* = 0.0007). While *N. campestris* showed a preference for *F. arundinacea*, no clear host preference was detected for *P. spumarius* (Figure 1). Additionally, the diversity of host species available also affected the oviposition behaviors (host diversity: F = 5.25; df = 1, 419; *p* = 0.0224), with this effect differing between the two vector species (vector × host diversity: F = 8.77; df = 1, 419; *p* = 0.0032). *Neophilaeunus campestris* maintained similar oviposition rates regardless of the host plant diversity, whereas *P. spumarius* had increased oviposition rates in scenarios with a single host species compared with those with multiple host species present, especially for *M. sativa* and, to a lesser degree, *C. arvensis* (Table 1).

#### Influence of Host Plant Diversity on Substrate Preference


*Neophilaenus campestris*


The choice of oviposition substrate by *N. campestris* (on-plant versus dry soil substrate) was influenced by the plant species (place × plant: F = 16.85; df = 2, 223; *p* < 0.0001) and the diversity of available host species (place × host diversity: F = 5.90; df = 1, 223; *p* = 0.016).

In the multi-host scenarios, the females laid eggs uniformly in the dry substrate surrounding all plant species. However, oviposition on the plant itself occurred exclusively on the preferred host: *F. arundinacea* (Figure 1A). When only one host species was available, substrate selection varied depending on the plant species (Figure 1B). (i) Oviposition was significantly higher on the plant substrate with *F. arundinacea*. (ii) Oviposition was significantly higher on the dry substrate, though some eggs were laid on the plant in *M. sativa*. (iii) Finally, oviposition occurred exclusively on the dry substrate with *C. arvensis*.


*Philaenus*
*spumarius*


In contrast, *P. spumarius* consistently preferred the dry soil substrate for oviposition, regardless of host diversity (place: F = 17.1; df = 1, 217; *p* < 0.0001; place × host diversity: F = 1.10; df = 2, 217; *p* = 0.2954). Eggs were only sporadically found on the plant substrate, primarily in *M. sativa* and occasionally in *C. arvensis* (Figure 1C,D).

### 3.2. Nymph Host Preferences

The distribution of *N. campestris* nymphs between the two host plant species varied significantly over time (time: F = 19.21; df = 1, 28; *p* = 0.0001). Although there was no overall preference for either plant species (host plant: F = 1.14; df = 1, 74; *p* = 0.2888), a strong interaction between the host plant and time was observed (host plant × time: F = 31.48; df = 1, 28; *p* < 0.0001). Fifteen minutes after their release, the nymphs were distributed evenly between the two host plant species (*p* = 0.7914). However, within 24 h, most nymphs had moved to *F. arundinacea* (*p* < 0.0001). By the eighth day, all *N. campestris* nymphs were exclusively found on *F. arundinacea* (Figure 2A).

In contrast, the *P. spumarius* nymphs showed a distinct and consistent preference for *M. sativa* from the start of the study (15 min after release), although a small proportion initially settled on *F. arundinacea*. Once they chose a host, the nymphs remained on that plant species throughout their development (host plant × time: F = 2.04; df = 1, 28; *p* = 0.1644). This behavior led to significant differences between the two host plant species across the study period (host plant: F = 60.64; df = 1, 224; *p* < 0.0001; time: F = 4.64; df = 1, 28; *p* = 0.040) (Figure 2B).

### 3.3. Nymph Survival and Development

#### 3.3.1. Nymph Survival

No significant differences were observed between the mean nymphal survival rates of *N. campestris* and *P. spumarius* when nymphs developed on their preferred host plants (F = 1.34; df = 1, 9; *p* = 0.2764; Table 2). *Neophilaenus campestris* nymphs successfully reached adulthood only when using *F. arundinacea* as their host plant. In contrast, the *P. spumarius* nymphs completed their development on *M. sativa*, *C. arvensis*, and *P. dulcis* but not on *F. arundinacea*. However, significant differences in survival rates were found across the three host plants that supported the development of *P. spumarius* nymphs (F = 17.54; df = 2, 8; *p* = 0.0012). Survival rates were lower on *P. dulcis* compared with *M. sativa* and *C. arvensis* (Table 2).

#### 3.3.2. Nymph Development

The developmental time for *P. spumarius* did not vary among plant species (F = 2.2; df = 2, 89; *p* = 0.116), (Table 3).

## 4. Discussion

Our study revealed distinct oviposition strategies between the two studied vector species (*N. campestris* and *P. spumarius*). As previously reported [34], *N. campestris* exhibited a strong preference for monocotyledonous plants, particularly *F. arundinacea*, over dicotyledonous species for oviposition sites. In contrast, while *P. spumarius* is often described as favoring dicotyledonous plants [25], this preference was less pronounced in our study. Host diversity had a significant impact on the *P. spumarius* oviposition behavior; females increased egg-laying on their preferential hosts, such as *M. sativa* (28 times more eggs per plant) or *C. arvensis* (3.5 times more eggs per plant), when only one host species was available. However, when multiple plant species were present, the oviposition rates decreased across all hosts, with no clear preferences for any particular species. A previous study by Morente et al. (2022) [35] examined *P. spumarius* oviposition preferences across seven plant species from four families and found significant variation in the oviposition rates among species. However, their study did not compare oviposition behavior in scenarios with a single host species versus multiple hosts. Vector females use both semiophysicals and semiochemicals for host selection. Visual cues are likely critical in host detection, while volatiles and xylem sap metabolites are key components of host acceptance. These responses vary with plant species and differ between vectors [36,37]. However, it remains unclear whether the presence of multiple plant species, each emitting different chemical signals, influences oviposition behavior. Further research in this area may help clarify the behavioral patterns observed in our work. Our findings suggest that increasing plant diversity in agricultural landscapes may reduce *P. spumarius* oviposition rates, even on its preferential hosts. While Gallego et al. (2023) [38] reported higher abundances of *P. spumarius* and *N. campestris* in less intensive agricultural areas with natural habitats, their study focused on regional-scale landscape factors rather than specific host plant availability. Local habitat manipulation, such as incorporating diverse cover crops and flower strips, could be a practical strategy to reduce *P. spumarius* colonization in agricultural fields.

The two species also exhibited distinct oviposition site preferences. Both laid eggs on plant substrates and dry soil substrates (e.g., pine needles), but their selection criteria were species-specific and mediated by host plant availability. The *Neophilaenus campestris* females oviposited directly on their preferred host, *F. arundinacea*, when this was the only available option. However, in multi-host scenarios, oviposition occurred equally on plant and dry substrates for *F. arundinacea*, while oviposition on dry substrates remained predominant for the non-preferential hosts. This behavior suggests a strong linkage between adult host preference and the developmental needs of their progeny, as females prioritize laying eggs on optimal hosts when possible. However, in the absence of such hosts, they opt for ground substrates, possibly anticipating that nymphs will relocate to suitable hosts after emergence. In agricultural systems dominated by grasses, such as *F. arundinacea* cover crops, winter mowing after the egg-laying period may help reduce nymph populations in the spring.

In contrast, the *P. spumarius* females predominantly laid eggs on dry soil substrates regardless of host diversity, with only marginal oviposition on preferential plants. This weak association between adult host preference and nymphal performance was also observed by Morente et al. (2022) [35], who found that some plants attracted oviposition but were unsuitable for nymphal development, while others were both attractive and suitable. Insect species exhibiting such weak preference-performance linkages are often more prone to population outbreaks [39]. Our findings align with previous studies showing that *P. spumarius* oviposits primarily on dry substrates, independent of host plant availability [25]. Interestingly, when a single host species was present, oviposition rates on dry soil substrates increased, while oviposition on plants remained minimal. Given this oviposition strategy, winter plant removal may be ineffective for controlling *P. spumarius*, as eggs are mainly deposited in soil substrates. Instead, spring mowing, as suggested by Bodino et al. (2021) [23], may be a more effective management approach.

A diversified agricultural landscape may further contribute to reducing *P. spumarius* oviposition, as oviposition rates on both plant and soil substrates were lower in the multi-host scenarios. The preference for soil substrates in *P. spumarius* may be linked to the ephemeral nature of many of its herbaceous hosts, which often lack green aerial parts during the egg-laying season [40]. Conversely, many *N. campestris* hosts, such as *F. arundinacea*, are perennial grasses with green aerial parts year-round.

Nymphal behavior also differed between the two species, aligning with the oviposition strategies of gravid females. The *Philaenus spumarius* nymphs rapidly located and settled on their preferred host, *M. sativa*, where they remained until adulthood. A small portion of the nymphs settled on a non-preferred host, *F. arundinacea*, and stayed there throughout development. This ability to locate preferential hosts, combined with the observed oviposition behavior of gravid females—who predominantly laid eggs on soil substrates in scenarios with a single preferential host species—suggests an adaptation to exploiting habitats where such hosts are abundant, even if unavailable at the time of oviposition.

In contrast, the *N. campestris* nymphs initially moved to both preferred and non-preferred hosts but abandoned the non-preferred hosts within a week, ultimately settling exclusively on *F. arundinacea*. This supports the idea that female oviposition decisions in *N. campestris* play a crucial role in offspring survival, although nymphs exhibit some capacity to relocate when placed on unsuitable hosts [41].

Despite their polyphagy, nymph survival was strongly influenced by the host plant species, with clear differences between the two vectors. The *Neophilaenus campestris* nymphs only developed on *F. arundinacea*, while the *P. spumarius* nymphs failed to survive on this host. Although previous studies have identified Poaceae species as potential hosts for *P. spumarius* [25], our results suggest that their role may be limited. Some species commonly used in agriculture cover crops, such as *F. arundinacea*, might function as alternative, non-preferential hosts for adult feeding when better options are unavailable. However, Poaceae species should not be disregarded in *P. spumarius* management strategies, as they may serve as temporary refuges when preferred hosts are scarce [42].

Although *P. spumarius* nymphs could develop on *P. dulcis*, the survival rates were significantly lower than on the preferred host. This finding confirms that *P. dulcis* primarily serves as a secondary food resource for adults when more suitable plant species are unavailable. Since *X. fastidiosa* vectors feed on xylem sap—a chemically simple fluid with low nutritional value and little variation across plant species [43]—differences in host plant preferences among xylem-feeding insects are likely driven by secondary defense metabolites [44] or specific anatomical traits rather than nutritional content.

The developmental time also varied between the two species, with the *N. campestris* nymphs reaching adulthood in half the time required by the *P. spumarius* nymphs. This likely explains the earlier seasonal appearance of *N. campestris* adults [19,45].

Finally, it is important to note that this study was conducted using *X. fastidiosa*-free insects. While the potential impact of bacterial infection on vector behavior remains unknown, field observations indicate that the vast majority of the adult females in the studied pathosystem were uninfected, and all recently emerged nymphs were free of *X. fastidiosa*. Future research should explore whether the infection status influences vector behavior and its implications for disease transmission dynamics.

## 5. Conclusions

The two primary European vectors of *X. fastidiosa*, *N. campestris*, and *P. spumarius* exhibited distinct strategies for oviposition and nymph development. Beyond host plant preferences, the diversity of plant species in oviposition habitats significantly influenced female oviposition behavior. Our findings suggest that habitat management could serve as a tool for mitigating the spread of *X. fastidiosa*. Strategies such as increasing plant diversity within agroecosystems and adjacent non-cropped areas, as well as winter mowing of grasses to eliminate *N. campestris* egg sites or spring mowing to remove *P. spumarius* nymphs before they develop into adults, could help manage vector populations and limit disease transmission.

## Figures and Tables

**Figure 1 insects-16-00416-f001:**
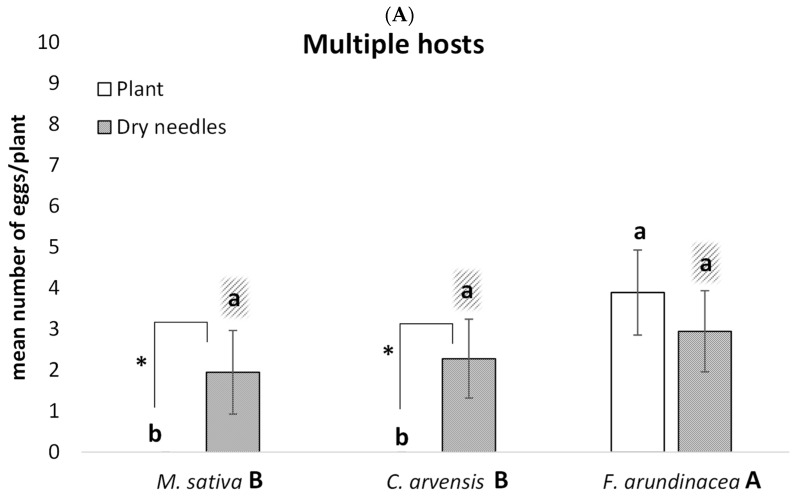
Mean number (±SE) of eggs laid per plant by adult *N. campestris* and *P. spumarius* females during seven days on three host plant species (*M. sativa*, *C. arvesnis*, and *F. arundinacea*) and in two host diversity scenarios. (**A**) *N. campestris* with the three plant species available in the same arena (multiple hosts). (**B**) *N. campestris* with only one plant species available in the same arena for oviposition (unique host). (**C**) *P. spumarius* with the three plant species available in the same arena (multiple hosts). (**D**) *P. spumarius* with only one plant species available in the same arena for oviposition (unique host). For each scenario, oviposition was measured on two types of substrates: (i) eggs laid on the host plant (plant substrate) and (ii) eggs laid on the soil dry substrate placed surrounding each plant (dry pine needles). For each scenario, different capital letters indicate significant differences in overall oviposition rates among plant species. Different lowercase letters indicate significant differences in oviposition rates among plant species within each oviposition substrate (plant or dry needles). Asterisks indicate significant differences in oviposition rates between the two substrates (plant vs. dry needles) for each plant species (Tukey post hoc test; *p* < 0.05).

**Figure 2 insects-16-00416-f002:**
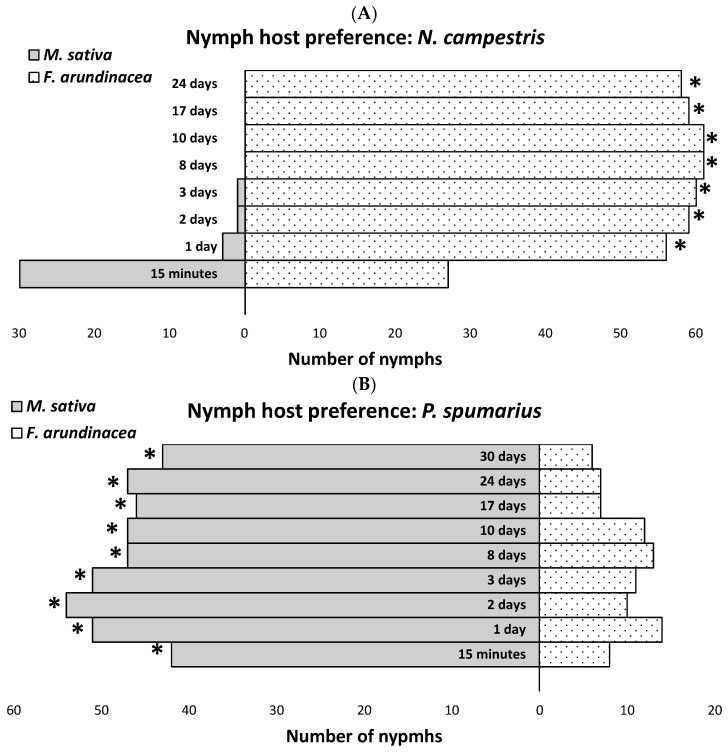
Number of (**A**) *N. campestris* and (**B**) *P. spumarius* nymphs found on either *M. sativa* or *F. arundinacea* at different times after their release onto a dry soil substrate. Asterisks indicate significant differences between host plant species at each time point (*p* < 0.05; exact test of goodness of fit).

**Table 1 insects-16-00416-t001:** Mean number (±SE) of eggs laid per plant by adult *N. campestris* and *P. spumarius* females during seven days on three host plant species (*M. sativa*, *C. arvensis*, and *F. arundinacea*) under two scenarios: (1) diverse host plant presence (three species available) and (2) monospecific host plant presence (one species available).

	Diverse Host Plant Presence				Monospecific Host Plant Presence			
	*M. sativa*	*C. arvensis*	*F. arundinacea*	Mean	*M. sativa*	*C. arvensis*	*F. arundinacea*	Mean
*N. campestris*	1.94 ± 1.02	2.28 ± 0.96 *	6.83 ± 1.4	**3.69 ± 0.72**	4 ± 1.32	0.52 ± 0.38	8.67 ± 1.59	**4.4 ± 0.81**
*P. spumarius*	0.18 ± 0.18	0.94 ± 0.69	1 ± 0.57	**0.71 ± 0.3**	5.1 ± 1.58 **	3.29 ± 1.37 *	1.38 ± 0.6	**3.25 ± 0.74 ****

For each vector species, differences in oviposition rates between host diversity scenarios (single vs. multiple host species) are indicated by two asterisks (*p* < 0.05) or one asterisk (*p* < 0.10). Comparisons were made for both the overall mean values across all plant species (in bold) and individually for each plant species.

**Table 2 insects-16-00416-t002:** Survival rates (%) for *N. campestris* and *P. spumarius* on four host plant species: *C. arvensis*, *M. sativa*, *P. dulcis*, and *F. arundinacea*. Preferred hosts are indicated (*F. arundinacea* for *N. campestris* and *C. arvensis* and *M. sativa* for *P. spumarius*).

	Survival (%)				
	Preferred Hosts	*C. arvensis*	*M. sativa*	*P. dulcis*	*F. arundinacea*
*N. campestris*	**75.15 ± 7.58**	0.00 ± 0.00b	0.00 ± 0.00b	0.00 ± 0.00b	75.15 ± 7.58a
*P. spumarius*	**83.18 ± 5.02**	82.41 ± 5.77a	82.41 ± 5.77a	27.61 ± 6.94b	0.00 ± 0.00c

Different lowercase letters denote significant differences between host plant species for each insect species (Tukey post hoc test; *p* < 0.05).

**Table 3 insects-16-00416-t003:** Nymphal developmental times (days) for *N. campestris* and *P. spumarius* on four host plant species: *C. arvensis*, *M. sativa*, *P. dulcis*, and *F. arundinacea*. Preferred hosts are indicated (*F. arundinacea* for *N. campestris* and *C. arvensis* and *M. sativa* for *P. spumarius*).

	Developmental Time (days)				
	Preferred Hosts	*C. arvensis*	*M. sativa*	*P. dulcis*	*F. arundinacea*
*N. campestris*	**28.21 ± 0.88**	0.00 ± 0.00b	0.00 ± 0.00b	0.00 ± 0.00b	28.21 ± 0.88a
*P. spumarius*	**50.08 ± 0.75**	48.37 ± 1.09a	51.61 ± 1.90a	50.64 ± 1.90a	0.00 ± 0.00b

Different lowercase letters denote significant differences between host plant species for each insect species (Tukey post hoc test; *p* < 0.05).

## Data Availability

Data are available from the IVIA repository.

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
