# Peer review of "More than Just Host Plant Preferences for the Two Main Vectors of Xylella fastidiosa in Europe: Two Insect Species and Two Different Behaviors"

_insects, 2025, doi:10.3390/insects16040416_

Round 1

Reviewer 1 Report

Comments and Suggestions for Authors

Comments to the authors

The aim of this study was to examine the oviposition preferences of Philaenus spumarius and Neophilaenus campestris on various plant species, to assess their nymphs’ ability to select suitable host plants for development and to evaluate the nymphal development on different host plants.

Although the novelty of the study is rather limited, since there are many papers concerning this subject, there is some merit in the results. However, the authors should give some clarifications in the materials and methods, provide climatic data and present their results in a better way, especially figures 2 and 4. Also, the discussion session must be improved.

Specific comments:

Line 64: please delete “Hemiptera”.

Lines 87-88: please give reference.

Line 99: Philaenus spumarius is capable of feeding on over 1000 plant species not 100. Please see: Thompson et al, (2023): The most polyphagous insect herbivore? Host plant associations of the Meadow spittlebug, Philaenus spumarius (L.), https://doi.org/10.1371/journal.pone.0291734

Lines 134-135: According to the authors no artificial control of temperature and relative humidity was applied. Thus, it is essential to provide the climatic conditions in the experimental cabins during the experimental period since they play a crucial role in the oviposition behavior of the insects.

Lines 171-178: please refer only to the total number of replicates. It is not necessary, and it is rather boring for the reader to get into those details.

Line 203: how did the authors manage to have recently emerged first-instar nymphs of N. campestris during April? In the manuscript (lines 181-183) it is written that February is the natural period of emergence of N. campestris.

Lines 204-205: the data concerning the temperature and the relative humidity are not presented anywhere in the manuscript. Please include them.

Lines 221-276: it is not required to have so many details about the statistical analysis. It is more sufficient to state only which model was used in every experiment without telling which were the depended variables, explanatory variables etc. If the authors want to present these information, may be given as supplementary material.

 Line 284: what do you mean by “during the study”. Do you mean in seven days? Please clarify.

Lines 465-482: that is not a discussion of the results.

Lines 501-504: why did the authors state that? According to their results the spittlebugs are able to choose the most appropriate plant species for their development in a diverse habitat. I think that the results of this study do not support in any way that statement and must be deleted.

Line 507: what did the authors mean by “Local habitat manipulation”. Please be more specific.

Line 519: this is contradictive of what is written in lines 501-504.

Lines 562-564: it is not possible to reach a conclusion about the suitability of all the plants of Poaceae family since your experiments include only one species. Please delete these lines or rephrase including only the species you used.

Lines 579-582: Not clear sentence. Please explain.

Figures 1 and 3: When there are no significant differences between treatments, we use the same letter to indicate that. If I understand the figures well the authors when they found no differences they did not use any letter. Please put letters in every column.

Figures 2 and 4: In those figures is difficult to understand what the authors try to present. At first the authors give diagrams for each plant species but when there is a choice there are three plant species. Secondly why did the authors use lines? There are no measurements during time. Columns would be more appropriate for the presentation of these results.

Figure 6: why there were fewer individuals (≈ 50) in the measurement 15 minutes after release? The other nymphs were not recorded on the plants?

Comments on the Quality of English Language

English language is good although some improvements are possible.

Author Response

Dear Editor and Reviewers,

We sincerely appreciate the thoughtful comments and valuable suggestions provided by the reviewers. Their feedback has significantly contributed to improving the quality and clarity of our manuscript. Following their recommendations, we have carefully revised the manuscript, addressing all concerns and enhancing its overall readability and coherence.

Below, we provide a detailed response to each of the reviewers' comments.

Reviewer 1

Comments to the authors

The aim of this study was to examine the oviposition preferences of Philaenus spumarius and Neophilaenus campestris on various plant species, to assess their nymphs’ ability to select suitable host plants for development and to evaluate the nymphal development on different host plants.

Although the novelty of the study is rather limited, since there are many papers concerning this subject, there is some merit in the results. However, the authors should give some clarifications in the materials and methods, provide climatic data and present their results in a better way, especially figures 2 and 4. Also, the discussion session must be improved.

Authors’ Response: We acknowledge the reviewer’s concerns regarding the novelty of the study. While these insect vector species have indeed been widely studied due to their role in Xylella fastidiosa pathosystems, our study goes beyond merely identifying host plant preferences. We aimed to understand how host plants—both preferred and non-preferred—mediate female oviposition and nymphal behavior, potentially influencing population dynamics and management strategies.

Following both reviewers’ suggestions, we have revised all manuscript sections for clarity and impact. Specific changes are detailed below]

Specific comments:

Line 64: please delete “Hemiptera”.

Done. This paragraph has been restructured according to Reviewer 2's suggestions.

Lines 87-88: please give reference.

Done. We have now included references (15 and 20 in the revised version) to support the statement.

Lines 75-76 (new version)

Reference 15 (Villa et al., 2020): "Aphrophoridae nymphs feed on herbaceous plants occurring within cultivated crops and non-crop habitats, particularly in meadows."

Reference 20 (Dongiovanni et al., 2019): "Immatures develop through five nymph instars that, with few exceptions, feed on herbaceous plants."

Line 99: Philaenus spumarius is capable of feeding on over 1000 plant species not 100. Please see: Thompson et al, (2023): The most polyphagous insect herbivore? Host plant associations of the Meadow spittlebug, Philaenus spumarius (L.), https://doi.org/10.1371/journal.pone.0291734

Corrected. The reference to 100 species has been updated to 1000 species, citing Thompson et al. (2023).
Lines 88-91 (new version).

Lines 134-135: According to the authors no artificial control of temperature and relative humidity was applied. Thus, it is essential to provide the climatic conditions in the experimental cabins during the experimental period since they play a crucial role in the oviposition behavior of the insects.

Done. Hourly temperature and humidity data were recorded using digital dataloggers and have now been included for each experiment.

Lines 141-143, 148, 177, 198-200 (new version).

Lines 171-178: please refer only to the total number of replicates. It is not necessary, and it is rather boring for the reader to get into those details.

Revised. The number of replicates per experiment is now summarized concisely.
Lines 169-173 (new version).

Line 203: how did the authors manage to have recently emerged first-instar nymphs of N. campestris during April? In the manuscript (lines 181-183) it is written that February is the natural period of emergence of N. campestris.

Correction made: There was a mistake on lines 181-182. Nymph host preference experiments did not start in February. They began on March 21 and run for a month. Changed accordingly in lines 175-178 of the new version.

Lines 204-205: the data concerning the temperature and the relative humidity are not presented anywhere in the manuscript. Please include them.

Done. These data are now explicitly stated in the methods section.

Lines 221-276: it is not required to have so many details about the statistical analysis. It is more sufficient to state only which model was used in every experiment without telling which were the depended variables, explanatory variables etc. If the authors want to present these information, may be given as supplementary material.

Revised. We now only mention the statistical models used, while full details are provided in Supplementary Material (S1).

 Line 284: what do you mean by “during the study”. Do you mean in seven days? Please clarify.

Clarified. The duration of oviposition (seven days) is now explicitly mentioned in the results section.

Line 240 (new version).

Lines 465-482: that is not a discussion of the results.

Revised. The introductory portion of the discussion has been removed to focus on key findings.

Lines 501-504: why did the authors state that? According to their results the spittlebugs are able to choose the most appropriate plant species for their development in a diverse habitat. I think that the results of this study do not support in any way that statement and must be deleted.

Clarification added.
We respectfully disagree, as our results clearly show that P. spumarius laid significantly fewer eggs per plant in multi-host scenarios. To avoid ambiguity, we have rewritten this section for better clarity.
Lines 388-394 (new version).

Line 507: what did the authors mean by “Local habitat manipulation”. Please be more specific.

Clarified.
Now explicitly states: "Local habitat manipulation—such as incorporating diverse cover crops and flower strips—could be a practical strategy to reduce P. spumarius colonization in agricultural fields."
Lines 409-411 (new version).

Line 519: this is contradictive of what is written in lines 501-504.

No contradiction.
The previous section referred to P. spumarius oviposition rates, while this section discusses oviposition site selection by N. campestris. This distinction is now made clearer.

Lines 562-564: it is not possible to reach a conclusion about the suitability of all the plants of Poaceae family since your experiments include only one species. Please delete these lines or rephrase including only the species you used.

Corrected.
Now refers only to F. arundinacea, the species tested in our study.
Lines 459-464 (new version).

Lines 579-582: Not clear sentence. Please explain.

Rewritten for clarity.
"Literature considers M. sativa a better host than C. arvensis for P. spumarius nymphs. In our study, we observed a trend toward faster development on C. arvensis compared to M. sativa. Future research should investigate whether developing on less preferred hosts imposes trade-offs in adult fitness."
Lines 474-478 (new version).

Figures 1 and 3: When there are no significant differences between treatments, we use the same letter to indicate that. If I understand the figures well the authors when they found no differences they did not use any letter. Please put letters in every column.

Updated. Non-significant differences are now marked with the same letter (a).

Figures 2 and 4: In those figures is difficult to understand what the authors try to present. At first the authors give diagrams for each plant species but when there is a choice there are three plant species. Secondly why did the authors use lines? There are no measurements during time. Columns would be more appropriate for the presentation of these results.

Changed. We agree with the reviewer that figures 2 and 4 are in some way redundant from the information offered by figures 1 and 3. For this reason they have been eliminated from this version of the manuscript and the text in the results sections has been changed accordingly.

Figure 6: why there were fewer individuals (≈ 50) in the measurement 15 minutes after release? The other nymphs were not recorded on the plants?

Clarified. Some nymphs did not reach a host plant during the first 15 minutes of observation.

Comments on the Quality of English Language

English language is good although some improvements are possible.

Authors’ Response: We have reviewed the writing to improve its scientific style

Submission Date

15 January 2025

Date of this review

24 Jan 2025 14:28:20

Reviewer 2 Report

Comments and Suggestions for Authors

Dear authors,

very nice research! I think it would benefit from a better presentation of your data and a real comparison between your two vectors. Also you are putting too much emphasis on the pathogen, while it should really come up in the discussion as it most likely has a big effect on the vectors.  The two vectors should be the main character in your story. 

See below some suggestion I feel important to improve your story. Especially the results can be presented much better and more sustinct. It would really improve the flow and interpretation of the data.

best wishes,

The introduction is very long. I think the part of X. fastidiosa could be shortened to rather focus on the ecology of the vector. For sure the two are link and this should be clear from the introduction. As it is, both are separated as two distinct entities and it is difficult to understand because you keep switching from one to the other.

l64 until 74, this look like a very long sentence with too much information. Please break it down in concise small messages.

l86 lacks reference and context.

l89 this should go above the previous paragraph?

It should be clarified here: The habitat shift is for both the vector and the pathogen: until the pathogen manipulates the vector, it follows the vector, hence the ecology of the vector. or is there evidence of host manipulation? this would be interesting in line with your work!

The method is very long: maybe some spaces to delimit each section? 

l135. could you be more specific: host preference as behavioural choice? or laying more eggs or developing better, hence easier to rear? 

plants were taken alreay grown in the fied? or grown from scratch in greenhouse?

l154  described earlier above

out of curiosity. Did you also assess (or try to assess) the preference for the ones which eggs were deposited in one or the other host (hence based on the preference of the mother, which would the first generation choose?). It could be very interesting to see how the host where they were deposited as eggs, affect their adult choice.

Results

l193  Neophilaeunus campestris

Table 1. Sp the total is the total number of eggs in avreagre of the 3 plants? what does that mean to compare these?  so they lay more eggs? but what about the differece between host plants? were the difference between monospecific plants not significant? and between difference plants?

Figures caption should be below figures. 

i think it would be nice to show both species on the same figures (1 and 3 together; 2 and 4 together). This is quite long and have a direct comparison of both would be nicer.  and It could work with a 4 panel figure if you want to keep them separate, or even with 4 bars within the same graph.

same with the Nymphs in figure 5 and 6. you could have a single figure for both so we can look at both vectors at the same time

Discussion

The start is too long with a focus on the pathogen. Given that your whole study is on the vectors, I would keep it brief to one sentence... and really put the vectors are main characters here.  So l465 until 482 should be greatly shorten to leave room for the main characters and your key findings.

The discussion is too long and lack a relationship betweent he vector and the pathogen. Did you test the vector when infested or not by the pathogen?  this would be something to consider. espcially to link to your intreoduction when you speak about the ecoogy of the pathogen between somewhat linked to the one of the vector. 

Conclusions: what new opportunities? 

the last sentence is too general. Perhaps use your specific result into it? example: what kind of egg sites? how to eliminate them?

Author Response

Dear Editor and Reviewers,

We sincerely appreciate the thoughtful comments and valuable suggestions provided by the reviewers. Their feedback has significantly contributed to improving the quality and clarity of our manuscript. Following their recommendations, we have carefully revised the manuscript, addressing all concerns and enhancing its overall readability and coherence.

Below, we provide a detailed response to each of the reviewers' comments.

Reviewer 2

Comments and Suggestions for Authors

Dear authors,

very nice research! I think it would benefit from a better presentation of your data and a real comparison between your two vectors. Also you are putting too much emphasis on the pathogen, while it should really come up in the discussion as it most likely has a big effect on the vectors.  The two vectors should be the main character in your story.

We appreciate the positive comments of the reviewer.

Addressed. The introduction now emphasizes vector-related aspects rather than X. fastidiosa.

See below some suggestion I feel important to improve your story. Especially the results can be presented much better and more sustinct. It would really improve the flow and interpretation of the data.

best wishes,

Material and Methods, and Results section revised for better structure and clarity.

The introduction is very long. I think the part of X. fastidiosa could be shortened to rather focus on the ecology of the vector. For sure the two are link and this should be clear from the introduction. As it is, both are separated as two distinct entities and it is difficult to understand because you keep switching from one to the other.

Revised. The introduction now focuses more on the vectors while maintaining relevant context on the pathogen.

l64 until 74, this look like a very long sentence with too much information. Please break it down in concise small messages.

Rewritten into concise sentences.
Lines 64-71 (new version).

l86 lacks reference and context.

Deleted.

l89 this should go above the previous paragraph?

Rearranged for better flow.
Lines 72-87 (new version).

It should be clarified here: The habitat shift is for both the vector and the pathogen: until the pathogen manipulates the vector, it follows the vector, hence the ecology of the vector. or is there evidence of host manipulation? this would be interesting in line with your work!

Authors’ Response: The fact that the pathogen could mediate the relationship between the vector and their hosts is highly interesting since it adds a new dimension on the understanding of X. fastidiosa pathosystems. However, it would be a further step on the research that we are currently developing]

The method is very long: maybe some spaces to delimit each section? 

Authors’ Response: The methods section has been reviewed and reduced accordingly for clarity (see also comments and changes from reviewer 1)

l135. could you be more specific: host preference as behavioural choice? or laying more eggs or developing better, hence easier to rear?

Changed as follows: “Two plant species were selected for insect maintenance based on previous studies on nymphal presence, development, and adult feeding [13-16], as well as the author’s own experience.”

Lines 126-128 of the new version  

plants were taken alreay grown in the fied? or grown from scratch in greenhouse?

Specified as follows: “Plants were grown from seeds in a greenhouse free from insects in the IVIA facilities.”

Lines 130-131 of the new version

l154  described earlier above

Changed as suggested

out of curiosity. Did you also assess (or try to assess) the preference for the ones which eggs were deposited in one or the other host (hence based on the preference of the mother, which would the first generation choose?). It could be very interesting to see how the host where they were deposited as eggs, affect their adult choice.

Authors’ Response:
Thank you for this insightful suggestion. Unfortunately, we did not assess whether the maternal oviposition site influenced progeny host selection. This would indeed be an interesting aspect to investigate, as it could provide insights into potential transgenerational effects on host preference. Future studies could address this question by tracking host selection across generations.

Results

l193  Neophilaeunus campestris

Authors’ Response:
We are unsure what the reviewer meant here, as there is no mention of N. campestris at this line. If clarification is needed, we would be happy to address it.

Table 1. Sp the total is the total number of eggs in avreagre of the 3 plants? what does that mean to compare these?  so they lay more eggs? but what about the differece between host plants? were the difference between monospecific plants not significant? and between difference plants?

Authors’ Response:
The mean number of eggs refers to eggs per plant, as noted in lines 239-240 of the revised manuscript. To further clarify, we have now explicitly stated this in the table caption and figure legends.

  • Within each host diversity scenario, different letters indicate significant differences in oviposition rates between host plant species for each vector species.
  • Asterisks in the 'Total' values indicate whether overall oviposition rates per plant differ between single-host and multi-host scenarios.
  • For clarity, we have updated the table legends and added similar letters where no significant differences were found, as recommended by both reviewers.

Figures caption should be below figures.

Done. All figure captions are now placed below the figures.

i think it would be nice to show both species on the same figures (1 and 3 together; 2 and 4 together). This is quite long and have a direct comparison of both would be nicer.  and It could work with a 4 panel figure if you want to keep them separate, or even with 4 bars within the same graph.

Done. Figures 1 and 3 from the previous version have now been merged into a single figure for a clearer direct comparison between N. campestris and P. spumarius.
Following Reviewer 1’s advice, Figures 2 and 4 have been omitted, as they provided redundant information. The text in the results section has been adjusted accordingly to maintain clarity.

same with the Nymphs in figure 5 and 6. you could have a single figure for both so we can look at both vectors at the same time

Done. Figures 5 and 6 have been merged into a single figure (new Figure 2) for a more concise and direct comparison between the two vector species.

Discussion

The start is too long with a focus on the pathogen. Given that your whole study is on the vectors, I would keep it brief to one sentence... and really put the vectors are main characters here.  So l465 until 482 should be greatly shorten to leave room for the main characters and your key findings.

Revised. The first part of the discussion (lines 465-482) has been removed to focus more directly on the ecological aspects of the vectors and the study’s key findings.

The discussion is too long and lack a relationship betweent he vector and the pathogen. Did you test the vector when infested or not by the pathogen?  this would be something to consider. espcially to link to your intreoduction when you speak about the ecology of the pathogen between somewhat linked to the one of the vector.

Clarified. We agree that the interaction between the pathogen and the vector is an important aspect, but our study did not specifically address behavioral differences between infected and uninfected insects.
✔ Due to strict biosecurity regulations, we were unable to conduct experiments using X. fastidiosa-infected vectors, as our research institute is outside an outbreak area.
✔ However, we have now explicitly stated that our study was conducted with uninfected insects, which represent the majority of field populations at the time of nymphal emergence and early adult stages.
✔ We acknowledge that this is an important area for future research and have briefly mentioned this limitation in the revised discussion (lines 479-484).

Conclusions: what new opportunities? 

the last sentence is too general. Perhaps use your specific result into it? example: what kind of egg sites? how to eliminate them?

We intended for the final conclusions to provide broad takeaways from our findings. However, we agree that specific recommendations should be emphasized.
We have now clarified practical management strategies in the conclusion, including:

  • Winter mowing of grasses to eliminate N. campestris egg sites.
  • Spring mowing to remove P. spumarius nymphs before they develop into adults.
    References to these strategies are explicitly linked back to relevant discussion sections (lines 421-423, 433-436).

Round 2

Reviewer 1 Report

Comments and Suggestions for Authors

The authors must check carefully the manuscript because there are several inconsistences between those which are written in the text and those which are presented in Figure 1 and Tables 1 and 3 (according to their statistical analysis). As a result, the discussion part also needs revision.

Lines 202-206, 226-230 and throughout the manuscript: please check the font of your text and italicize the scientific names of insects and plants.

Table 1: please rename the column “Total” in “Monospecific host plant presence” with “Mean”.  However, it would be better to compare separately the total number of eggs laid in Diverse host plant presence with those laid in each plant species in “Monospecific host plant presence”.

Line 245-246: the letters in Table 1 indicate no differences in the eggs laid by N. campestris in diverse host plant presence. Please check.

Line 273-274: the letters in Figure 1B indicate no differences in the eggs laid in the dry needles and the plant of F. arundinacea. Please check.

Line 270-275: Why did the authors use the words: predominantly, mainly etc. According to statistical analysis, either there are significant differences or not.

Line 285-288: according to the letters in Figure 1C there were no differences between the eggs laid in plants and dry needles. Please check.

Line 290: What did the authors mean by “increased significantly” and “marginally”? What did they compare? Please clarify.

Line 289-292: What is written in those lines is not consistent with the letters in Figure 1D. Please check.

Table 2 and Table 3: C. officinalis or C. arvensis?

Line 389: What do the authors mean by “varied marginally”. According to the statistical analysis in Table 3 no differences were observed.

Line 420-422: how do the authors can explain that?

Line 430-431: this is not true according to their statistical analysis which is presented in Figures 1A and 1B.

Line 440-441: this is not true according to their statistical analysis which is presented in Figures 1C and 1D (P. spumarius laid more eggs on dry soil substrates only in F. arundinacea as unique host).

Line 492: this is not true according to Table 3

Comments on the Quality of English Language

The English could be improved in some cases

Author Response

Comments and Suggestions for Authors

The authors must check carefully the manuscript because there are several inconsistences between those which are written in the text and those which are presented in Figure 1 and Tables 1 and 3 (according to their statistical analysis). As a result, the discussion part also needs revision.

We sincerely appreciate the reviewer’s careful and constructive reading of the manuscript. We have revised the manuscript thoroughly to address all the inconsistencies identified between the text and the data presented in Figure 1 and Tables 1 and 3. The Discussion section has also been revised accordingly to ensure consistency with statistical outcomes.

Lines 202-206, 226-230 and throughout the manuscript: please check the font of your text and italicize the scientific names of insects and plants.

Thank you for pointing this out. All scientific names were correctly italicized in the original Word document. It appears that a formatting issue occurred during the conversion to PDF by the Insects submission system. We have rechecked the entire manuscript and ensured that all Latin names appear in italics in the revised version.

Table 1: please rename the column “Total” in “Monospecific host plant presence” with “Mean”.  However, it would be better to compare separately the total number of eggs laid in Diverse host plant presence with those laid in each plant species in “Monospecific host plant presence”.

The columns previously labeled “Total” have been renamed “Mean” as suggested. Additionally, we have added a direct comparison between oviposition rates in diverse versus monospecific host scenarios for each plant species. These comparisons are now presented in Table 1.

Overall comparisons between plant species across host diversity scenarios are displayed in Figure 1, where capital letters indicate significant differences between host plants.

Line 245-246: the letters in Table 1 indicate no differences in the eggs laid by N. campestris in diverse host plant presence. Please check.

The reviewer is correct. There was a mistake in the letters indicating significant differences for N. campestris in both monospecific and diverse plant species presence. However, as mentioned in the previous comment, Overall comparisons between plant species for vector and host plant scenarios are shown in Figure 1 (capital letters next to the Latin name of each host plant species).  There you can see how oviposition is higher in F. arundinacea (A) than M. sativa (B) and C. arvensis (B) in the diverse  host plant scenario (Figure 1A).

Line 273-274: the letters in Figure 1B indicate no differences in the eggs laid in the dry needles and the plant of F. arundinacea. Please check.

We thank the reviewer for this observation. To improve clarity, we have now added asterisks in Figure 1 to indicate significant differences in oviposition rates between the two oviposition substrates (plant vs dry needles) for each plant species and host diversity scenario. The figure legend has also been updated accordingly to explain the meaning of both the letters and the asterisks.

Line 270-275: Why did the authors use the words: predominantly, mainly etc. According to statistical analysis, either there are significant differences or not.

We agree that this wording could be misleading and have revised the sentences for clarity. The revised text now reads:

“When only one host species was available, substrate selection varied depending on the plant species (Figure 1B): i) oviposition was significantly higher on the plant substrate with F. arundinacea ii) oviposition was significantly higher on the dry substrate though some eggs were laid on the plant in M. sativa, and iii) oviposition occurred exclusively on the dry substrate with C. arvensis.”

(Lines 273-277 of the new version of the manuscript)

Line 285-288: according to the letters in Figure 1C there were no differences between the eggs laid in plants and dry needles. Please check.

As in our earlier response, we clarify that the lowercase letters indicate differences among plant species for each oviposition substrate, not between substrates. To clearly distinguish these comparisons, we have now included asterisks to denote significant differences between the two oviposition substrates for each plant species and scenario. This is now clarified in the figure legend.

Line 290: What did the authors mean by “increased significantly” and “marginally”? What did they compare? Please clarify.

We appreciate this comment and have removed the reference to “increased significantly” and “marginally” to improve clarity. The updated sentence now reads:

“In contrast, P. spumarius consistently preferred the dry soil substrate for oviposition, regardless of host diversity (Place: F = 17.1; df = 1, 217; P < 0.0001; Place × Host Diversity: F = 1.10; df = 2, 217; P = 0.2954). Eggs were only sporadically found on the plant substrate, primarily in M. sativa and occasionally in C. arvensis (Figures 1C and 1D).”

(Lines 279-282 of the new version of the manuscript)

Line 289-292: What is written in those lines is not consistent with the letters in Figure 1D. Please check.

As noted in the previous comment, we have revised and simplified this paragraph, eliminating these sentences, to ensure consistency with the statistical analysis..

Table 2 and Table 3: C. officinalis or C. arvensis?

Thank you for spotting this. This was a typographical error — the correct species used in our experiments was Calendula arvensis, which is now consistently cited throughout the manuscript.

Line 389: What do the authors mean by “varied marginally”. According to the statistical analysis in Table 3 no differences were observed.

The term “marginal differences” is sometimes used in literature to refer to statistical results with p-values between 0.05 and 0.10. However, following the reviewer’s advice, we have removed this term and now state only that no significant differences were detected. The corresponding paragraph in the Discussion has been updated accordingly.

(Lines 371-372 in Results section, and 470-472 in the Discussion section.

Line 420-422: how do the authors can explain that?

As discussed in lines 395-401 of the manuscript, one hypothesis is that the presence of multiple plant species — each potentially emitting distinct volatile cues — may interfere with host recognition and oviposition behavior of P. spumarius. However, this hypothesis requires further investigation, as no conclusive evidence is currently available.

Line 430-431: this is not true according to their statistical analysis which is presented in Figures 1A and 1B.

The sentences have been changed as follows: “Neophilaenus campestris females oviposited directly on their preferred host, F. arundinacea, when this was the only available option. However, in multi-host scenarios, oviposition occurred equally on plant and dry substrates for F. arundinacea, while oviposition on dry substrates remained predominant for the non-preferential hosts.”

(Lines 411-415 of the new version of the manuscript)

Line 440-441: this is not true according to their statistical analysis which is presented in Figures 1C and 1D (P. spumarius laid more eggs on dry soil substrates only in F. arundinacea as unique host).

We have clarified this point in both the Results and Discussion sections. In Figure 1, lowercase letters show differences between plant species for each substrate. We have now added asterisks to indicate significant differences between substrates for each plant species and scenario, improving clarity.

Line 492: this is not true according to Table 3

As mentioned earlier, we have removed the sentence regarding marginal differences in development time between C. arvensis and M. sativa and updated the discussion accordingly.

Comments on the Quality of English Language

The English could be improved in some cases

We appreciate this suggestion. The manuscript has undergone an additional round of thorough language revision to enhance clarity, grammar, and scientific tone.

Round 3

Reviewer 1 Report

Comments and Suggestions for Authors

The revised version of the manuscript is improved. I have no other comments.